# Consumption of single products versus fixed-dose combination medicines for hypertension and hyperlipidemia during 2015–2019 in South Korea

**Yujin Lee**[ID]**, Jihye Shin, Yujeong Kim, Dong-Sook Kim**[ID]*

Department of Research, Health Insurance Review & Assessment Service (HIRA), Wonju, South Korea

* sttone@hanmail.net

## Abstract

### Background

Fixed-dose combinations can simplify prescribing, and numerous combination products exist for hypertension and dyslipidemia in South Korea. This study's aim was to compare trends in the consumption of single products versus fixed-dose combinations for hypertension and hyperlipidemia.

### Methods and findings

We analyzed the Korean national health insurance claims database from January 2015 through December 2019. Consumption of medicines was calculated using the defined daily dose per 1,000 inhabitants per day (DIDs) and expenditures over time. During 2015–2019, the use of antihypertensive drugs increased with an annual growth rate (AGR) of 0.9% for single products and with an AGR of 35.6% for fixed-dose combinations. A notable increase was observed for antihyperlipidemic combination drugs with an AGR of 268.1% compared to single products with 35.7%. For older adults (65+ years), the consumption of drugs for hypertension and hyperlipidemia was 3–4.5 and about 3 times higher, respectively, than in adults aged 20–64 years, and a sharp increase was found in antihyperlipidemic fixed-dose combinations among older adults. A large increase was seen for C09 (agents acting on the renin-angiotensin system) with an AGR of 36.5%, especially C09DB (angiotensin II receptor blockers + calcium channel blockers) was widely used and steeply increased with 114.2%. For antihyperlipidemic drugs, C10AA (HMG CoA reductase inhibitors) accounted for a large share and sharply increased, with 52.1 DIDs in 2019 and with an AGR of 78.4%, whereas C10BA (combinations of various lipid modifying agents) increased 9.6 times from 2.9 DIDs (96 million USD) in 2015 to 27.7 DIDs (912 million USD) in 2019.

**Data Availability Statement:** Data are available from the HIRA Institutional Data Access. (https://opendata.hira.or.kr/op/oph/selectCnfcUseAplPrsnt.do)".

**Funding:** The authors received no specific funding for this work.

**Competing interests:** The authors have declared that no competing interests exist.

## Conclusion

The findings of increased consumption and drug spending among older adults underscores the need for real-world evidence about health outcomes of fixed-dose combinations in this population.

## Introduction

Hypertension and dyslipidemia are known to be major risk factors for cardiovascular disease (CVD) [1]. In 36 Organization for Economic Co-operation and Development (OECD) countries, cardiovascular disease is the top cause of death as of 2017, accounting for 31% of deaths [2]. Therefore, to control blood pressure or cholesterol, the International Society has published treatment recommendations such as JNC 8. In South Korea (hereafter, Korea) in 2018, the prevalence of hypertension was 28.3%, that of hypercholesterolemia was 21.4%, and that of hypertriglyceridemia was 17.1% [3].

Fixed-dose combinations (FDCs) for antihypertensive or antihyperlipidemic drugs can make it easier for patients to take those drugs, thereby improving medication compliance. According to the National Committee (JNC 8) guideline in 2014, American College of Cardiology (ACC) and American Heart Association (AHA) guideline in 2017, and European Society of Cardiology/European Society of Hypertension guideline in 2018, FDCs can improve adherence and help to achieve treatment outcomes [1, 4–6]. Furthermore, the ACC/AHA guideline on the prevention of cardiovascular disease and the Clinical Practice Guidelines of the Korean Society of Lipid and Atherosclerosis recommended that combination therapy could lower low-density lipoprotein cholesterol levels [7, 8].

In Korea, 996 and 300 combination drugs for hypertension and dyslipidemia, respectively, were reimbursed in 2019, and the use of combination drugs is high [9], whereas most other countries generally use single-ingredient products. The OECD has published Health Statistics and Pharmaceutical Market Statistics, and requires each member country to submit pharmaceutical consumption data for drugs that have defined daily dose (DDD) values according to the World Health Organization (WHO) ATC-DDD classification [10]. The consumption of cardiovascular drugs such as antihypertensive and antihyperlipidemic drugs in Korea has been reported to be lower than that of other countries, while their expenditures are relatively high [11].

Several studies have analyzed the overall consumption of antihypertensive drugs [12–15]. However, few studies have examined single pill versus FDC therapy, because there are limited real-world data for comparisons of FDC use [16, 17]. Moreover, few studies analyzed use of antihyperlipidemic drugs [18]. Nationwide trends in the use of antihypertensive and antihyperlipidemic FDCs can provide insight into the need for further studies. Therefore, we aimed to examine patterns of use of combination drugs for hypertension and hyperlipidemia in Korea by patient age and therapeutic group using a nationwide database.

## Materials and methods

### Data source

We conducted a retrospective population-based study using the Korean national health insurance claims database from January 2015 through December 2019. This database contains information on both in-hospital and outpatient visits from a population of 51.8 million as of

2020. The database includes demographic characteristics, diagnosis, health care utilization, and medicine use (product name, ingredient name, dose, days of therapy, and spending). The analytical unit of this study was the medications listed in the health care claims data from the entire population.

## Study medicines and therapeutic class

Based on the WHO Anatomical Therapeutic Chemical (ATC) classification, drugs were selected if their ATC-2 classification indicated that they were antihypertensive drugs (C02, C03, C07, C08, C09) or antihyperlipidemic drugs (C10) [19]. For combination drugs, each drug was calculated by summing the DDDs for the main active ingredients. We classified the medicines into the ATC-4 level. The ATC-3 level refers to the pharmacological subgroup, and the ATC-4 level refers to the chemical subgroup. For example, level 3 C10A contains plain lipid-modifying agents, and level 4 C10AA refers to HMG CoA reductase inhibitors, and level 5 C10AA01 is simvastatin.

The number of combination drugs for hypertension increased from 767 in 2015 to 996 in 2019, and the number of combination drugs for hyperlipidemia steadily increased from 111 in 2015 to 300 in 2019. However, it temporarily decreased due to the valsartan carcinogen incident in 2019 [20].

## Measures and calculations

We analyzed changes in consumption and pharmaceutical spending according to each dimension using the health insurance database. The measure indicators were consumption and sales. Drug consumption figures were presented as numbers of DDDs/1000 inhabitants/day (DIDs), which is calculated as number of DDDs × 1000 / total population / 365, and pharmaceutical sales are given in units of millions of dollars [10, 11]. Using the ATC/DDD system allows standardization of drug groupings and stable drug utilization metrics to enable comparisons of drug use between countries, regions, and other health care settings, as well as making it possible to examine trends in drug use over time and in different settings. DIDs may provide a rough estimate of the proportion of the population within a defined area treated daily with certain drugs. DDDs provide a fixed unit of measurement independent of price, currency, package size, and strength, enabling researchers to assess trends in drug consumption and to perform comparisons between population groups. Pharmaceutical spending referred to total spending including value-added tax (VAT). We analyzed the consumption and spending according to each dimension.

The analytical dimensions were patients' age and drug classification based on (ATC) system. We used the number of population by year and age of the Statistic Korea. Sales were calculated in millions of dollars. The age of the patients was divided into those under 65 and over 65 years of age.

## Results

### Consumption and sale by single pill versus fixed-dose combinations

Table 1 examines the consumption and total pharmaceutical expenditures by categories of medicines. In 2019, the number of patients prescribed combination drugs for hypertension and hyperlipidemia was about 6 million and 2.3 million, respectively. During 2015–2019, the use of antihypertensive drugs increased from 123.4 DIDs to 124.9 DIDs with an annual growth rate (AGR) of 0.9% in single products and from 113.7 DIDs to 154.2 DIDs with 35.6% in FDCs. The antihypertensive FDC market share based on consumption increased from 48% in

**Table 1. Use and pharmaceutical expenditures on antihypertensive and antihyperlipidemic drugs.**

| | 2015 | | 2016 | 2017 | 2018 | 2019 | | Annual growth rate (%) |
|---|---|---|---|---|---|---|---|---|
| **Antihypertensive drugs** | | | | | | | | |
| Total use (DID) | 237.1 | | 247.2 | 256.6 | 268.2 | 278.7 | | (17.5) |
| Use of plain products (DID) | 123.4 | (52.0%) | 123.1 | 121.9 | 122.3 | 124.5 | (44.7%) | (0.9) |
| Use of FDC (DID) | 113.7 | (48.0%) | 124.1 | 134.7 | 145.9 | 154.2 | (55.3%) | (35.6) |
| Total expenditures (million USD) | 1,326 | | 1,381 | 1,431 | 1,506 | 1,569 | | (18.4) |
| Expenditures for plain products (million USD) | 736 | (55.5%) | 740 | 744 | 760 | 784 | (49.9%) | (6.5) |
| Expenditures for FDCs (million USD) | 590 | (44.5%) | 641 | 688 | 746 | 786 | (50.1%) | (33.2) |
| **Antihyperlipidemic drugs** | | | | | | | | |
| Total use (DID) | 71.3 | | 86.4 | 100.9 | 114.3 | 129.5 | | (81.6) |
| Use of plain products (DID) | 57.2 | (80.2%) | 63.4 | 67.6 | 72.7 | 77.6 | (59.9%) | (35.7) |
| Use of FDCs (DID) | 14.1 | (19.8%) | 23.0 | 33.3 | 41.6 | 51.9 | (40.1%) | (268.1) |
| Total expenditures (million USD) | 819 | | 943 | 1,075 | 1,200 | 1,347 | | (64.4) |
| Expenditures for plain products (million USD) | 688 | (83.9%) | 743 | 785 | 836 | 888 | (65.9%) | (29.2) |
| Expenditures for FDCs (million USD) | 132 | (16.1%) | 201 | 291 | 364 | 459 | (34.1%) | (248.4) |

DID: DDD/1,000 inhabitants/day

FDC: fixed-dose combinations.

2015 to 55.3% in 2019. Pharmaceutical expenditures for single products and FDC drugs increased from 0.7 billion USD in 2015 to 0.78 billion USD in 2019 and from 0.59 billion USD to 0.78 billion USD, respectively.

A notable increase was observed for antihyperlipidemic FDCs (from 14.1 DIDs and 0.1 billion USD in 2015 to 51.9 DIDs and 0.46 billion USD in 2019) with an AGR of 268.1% compared to single products (from 57.2 DIDs to 77.6 DIDs, and from 0.69 billion USD to 0.89 billion USD).

Table 2 shows trends in consumption and pharmaceutical expenditures by age groups. In older adults, the consumption of drugs for hypertension was 3–4.5 times higher than among adults 20–64 years of age, and the consumption of antihyperlipidemic drugs was about 3 times higher.

## Use by drug classification

Table 3 shows trends in antihypertensive drug use by ATC level 2 and level 3 classification. The consumption of C03 (diuretics) showed no change, that of C07 (beta-blocking agents) increased slightly, and that of C08 (calcium channel blockers) decreased slightly. The only obvious increase was found for C09 (agents acting on the renin-angiotensin system) with an AGR of 36.5%. C09D (ARBs, combination) increased greatly from 112.3 DIDs in 2015 to 153.3 DIDs in 2019 and C09C (ARBs, plain) increased from 40.1 DIDs to 45.8 DIDs, while C01A (ACE inhibitors plain) decreased from 5.6 DIDs to 4.1 DIDs.

Most drug spending increased except for C09A (ACE inhibitors, plain) and C09B (ACE inhibitors, combination). The greatest increase was noted in C09D (ARBs, combination), followed by C09C (ARBs, plain).

Table 4 shows trends by ATC level 4 drug classification. Among antihypertensive drugs, a large share of the market and increased consumption were found for C09DB (ARBs + CCBs), from 42.4 DIDs (0.6 billion USD) in 2015 to 90.8 DIDs (1.2 billion USD) in 2019 with an AGR of 114.2%, followed by C09DA (ARBs + diuretics), C08CA (CCBs, dihydropyridine

**Table 2. Use and spending on antihypertensive and antihyperlipidemic drugs by age.**

| | 2015 | 2016 | 2017 | 2018 | 2019 |
|---|---|---|---|---|---|
| **Antihypertensive drug use (DIDs)** | | | | | |
| Plain products in 20–64 years | 83.5 | 82.8 | 80.3 | 80.2 | 103.9 |
| Plain products in ≥65 years | 518.2 | 503.8 | 490.4 | 476.2 | 463.4 |
| FDCs in 20–64 years | 94.2 | 102.8 | 110.7 | 119.4 | 161.5 |
| FDCs in ≥65 years | 389.1 | 411.3 | 437.3 | 457.5 | 464.2 |
| **Antihypertensive drug spending (USD per capita)** | | | | | |
| Plain products in 20–64 years | 10.1 | 10.1 | 10.0 | 10.1 | 13.3 |
| Plain products in ≥65 years | 58.5 | 56.9 | 56.0 | 55.2 | 54.3 |
| FDCs in 20–64 years | 9.6 | 10.4 | 11.0 | 11.8 | 15.9 |
| FDCs in ≥65 years | 39.3 | 41.3 | 43.4 | 45.3 | 45.7 |
| **Antihyperlipidemic drug use (DIDs)** | | | | | |
| Plain products in 20–64 years | 46.2 | 50.8 | 53.1 | 56.2 | 76.1 |
| Plain products in ≥65 years | 201.4 | 218.6 | 231.6 | 243.6 | 251.5 |
| FDCs in 20–64 years | 11.1 | 19.1 | 28.2 | 35.0 | 55.8 |
| FDCs in ≥65 years | 51.0 | 75.4 | 104.4 | 125.7 | 151.0 |
| **Antihyperlipidemic drug spending (USD per capita)** | | | | | |
| Plain products in 20–64 years | 10.7 | 11.4 | 11.7 | 12.2 | 16.4 |
| Plain products in ≥65 years | 48.5 | 51.1 | 53.7 | 55.7 | 57.0 |
| FDCs in 20–64 years | 2.1 | 3.3 | 4.8 | 6.0 | 9.6 |
| FDCs in ≥65 years | 9.2 | 12.7 | 17.5 | 21.1 | 25.4 |

DID: DDD/1,000 inhabitants/day

FDC: fixed-dose combinations

Pharmaceutical expenditures per capita were calculated by dividing expenditures by the total population.

derivatives), and C09CA (ARBs, plain) which were single products. The highest pharmaceutical expenditures were found for C09DB (ARBs + CCBs), followed by C09DA (ARBs + diuretics), C09CA (ARBs, plain), and C08CA (CCBs, dihydropyridine derivatives).

For antihyperlipidemic drugs, C10AA (HMG CoA reductase inhibitors) accounted for a large share, with 52.1 DIDs and 1.6 billion USD in 2019, whereas C10BA (combinations of various lipid-modifying agents) increased 9.6 times from 2.9 DIDs (96 million USD) in 2015 to 27.7 DIDs (0.9 billion USD) in 2019, followed by C10BX (lipid-modifying agents in combination with other drugs) with 18.8 DIDs and 0.4 billion USD in 2019 (Table 4).

The ingredients that accounted for large amounts of antihypertensive drug consumption in 2019 were carvedilol (4.8 DIDs, 67 million USD), nebivolol (2.6 DIDs, 9 million USD), and bisoprolol (2.4 DIDs, 18 million USD) among beta-blockers; amlodipine (32.6 DIDs, 0.19 billion USD), nifedipine (4.1 DIDs, 18 million USD), diltiazem (2.6 DIDs, 24 million USD) among CCBs; and telmisartan+amlodipine (34.4 DIDs, 0.18 billion USD), valsartan+amlodipine (34.2 DIDs, 0.19 billion USD), olmesartan medoxomil+amlodipine (16.2 DIDs, 75 million USD), losartan+amlodipine (13.3 DIDs, 75 million USD), losartan +diuretics (12.8 DIDs, 60 million USD), and losartan (11.9 DIDs and 86 million USD). The ingredient with the largest consumption among antihyperlipidemic drugs was atovastatin (29.4 DIDs, 0.47 billion USD), followed by rosuvastatin (29.2 DIDs, 0.25 billion USD), rosuvastatin + ezetimibe (25.9 DIDs, 0.24 billion USD), pitavastatin (9.1 DIDs, 73.5 million USD) (Fig 1).

**Table 3.  Use and pharmaceutical expenditures for antihypertensive drugs by therapeutic class.**

| | 2015 | 2016 | 2017 | 2018 | 2019 | Annual growth rate (%) |
|---|---|---|---|---|---|---|
| **Antihypertensive drug use (DIDs)** | | | | | | |
| C02 (Antihypertensive) | 2.0 | 1.9 | 1.8 | 1.8 | 1.7 | (-11.1) |
| C03 (Diuretics) | 14.7 | 14.6 | 14.5 | 14.5 | 14.6 | (-0.5) |
| C07 (Beta-blocking agents) | 13.6 | 13.7 | 13.6 | 13.8 | 14.0 | (3.3) |
| C08 (Calcium channel blockers) | 48.6 | 47.7 | 46.4 | 45.4 | 44.7 | (-8.0) |
| C09 (Agents acting on the renin-angiotensin system) | 158.3 | 169.4 | 180.3 | 192.8 | 203.4 | (28.5) |
| • C09A (ACE inhibitors, plain) | 5.6 | 5.2 | 4.8 | 4.4 | 4.1 | (-26.6) |
| • C09B (ACE inhibitors, combination) | 0.3 | 0.3 | 0.3 | 0.2 | 0.2 | (-36.8) |
| • C09C (ARBs, plain) | 40.1 | 41.1 | 41.6 | 43.2 | 45.8 | (14.2) |
| • C09D (ARBs, combination) | 112.3 | 122.9 | 133.6 | 144.9 | 153.3 | (36.5) |
| **Antihypertensive drug spending (million USD)** | | | | | | |
| C02 (Antihypertensive) | 23 | 25 | 27 | 29 | 31 | (36.9) |
| C03 (Diuretics) | 16 | 16 | 16 | 17 | 19 | (19.3) |
| C07 (Beta-blocking agents) | 100 | 100 | 102 | 106 | 111 | (10.5) |
| C08 (Calcium channel blockers) | 284 | 279 | 274 | 271 | 270 | (-4.9) |
| C09 (Agents acting on the renin-angiotensin system) | 903 | 961 | 1,011 | 1,082 | 1,139 | (26.1) |
| • C09A (ACE inhibitors, plain) | 32 | 29 | 27 | 26 | 24 | (-25.9) |
| • C09B (ACE inhibitors, combination) | 2 | 2 | 2 | 1 | 1 | (-36.2) |
| • C09C (ARBs, plain) | 284 | 293 | 299 | 312 | 331 | (16.5) |
| • C09D (ARBs, combination) | 585 | 637 | 684 | 743 | 783 | (33.8) |

DID: DDD/1,000 inhabitants/day.

## Discussion

This study investigated national trends in antihypertensive and antihyperlipidemic drugs in terms of single products versus FDCs during 2015–2019 using national health insurance claims data. Of note, the patterns that we found are meaningfully different from those observed in other countries. During the past 5 years, ARBs and CCBs were the most commonly used drugs in Korea. In contrast, a study in Denmark found that ACE inhibitors were the most frequently used drugs. According to Sundbøll et al., the overall consumption of antihypertensive drugs was 379 DIDs, and the most commonly used therapeutic classes were ACE inhibitors with 105 DIDs, CCBs with 82 DIDs, diuretics with 81 DIDs, and ARBs with 72 DIDs [15]. Another study from Croatia stated that the most common ATC level 2 subgroup was C09 (agents acting on the renin-angiotensin system (RAS)), with 202 DIDs; followed by C08 (CCB), with 70.7 DIDs; and C03 (diuretics), with 48.2 DIDs in 2016 [13]. However, that study did not analyze more specific therapeutic groups. According to Huang et al., the highest utilization of antihypertensive drugs in Taiwan was found for CCBs (35.1 DIDs), followed by ACE inhibitors (19.6 DIDs) and beta-blockers (19.9 DIDs) [12]. A study from the Russian Federation analyzed the use of combination drugs and reported that the leading therapeutic class was the combination of ARBs + diuretics [17]. Similarly, a study in Campania (Italy) and Aragon (Spain) showed that ACE inhibitors were most frequently used, followed by ARBs and CCB or ARB + diuretics (40.2 DIDs, 13% in Campania and 33.9 DIDs, 13.2% in Aragon, respectively) [16]. However, notably, the combination of ARBs + CCBs was the most frequently used drug in Korea. We also found an increasing trend in this combination among older adults.

**Table 4. Use and spending on antihypertensive and antihyperlipidemic drugs by ATC class and single versus combination.**

| ATC class | | Drug use (DIDs) | | | | | Drugs spending (million USD) | | | | |
|---|---|---|---|---|---|---|---|---|---|---|---|
| | | 2015 | 2016 | 2017 | 2018 | 2019 | 2015 | 2016 | 2017 | 2018 | 2019 |
| **Antihypertensive drugs, plain** | | | | | | | | | | | |
| C02AC | Imidazoline receptor agonists | 0.0 | 0.0 | 0.0 | 0.0 | 0.0 | 0 | 0 | 0 | 0 | 0 |
| C02CA | Alpha-adrenoreceptor antagonists | 2.0 | 2.2 | 2.2 | 2.3 | 2.5 | 7 | 6 | 6 | 6 | 6 |
| C02DB | Hydrazinophthalazine derivatives | 0.0 | 0.0 | 0.0 | 0.0 | 0.0 | 0 | 0 | 0 | 0 | 0 |
| C02DC | Pyrimidine derivatives | 0.0 | 0.0 | 0.0 | 0.0 | 0.0 | 0 | 0 | 0 | 0 | 0 |
| C02DD | Nitroferricyanide derivatives | 0.0 | 0.0 | 0.0 | 0.0 | 0.0 | 0 | 0 | 0 | 0 | 0 |
| C02KX | Antihypertensives for pulmonary arterial hypertension | 0.0 | 0.0 | 0.0 | 0.0 | 0.0 | 35 | 43 | 55 | 62 | 67 |
| C03AA | Thiazides, plain | 2.7 | 2.6 | 2.5 | 2.6 | 2.5 | 0 | 0 | 0 | 0 | 0 |
| C03BA | Sulfonamides, plain | 1.1 | 1.0 | 1.1 | 1.2 | 1.6 | 8 | 6 | 7 | 7 | 8 |
| C03CA | Sulfonamides, plain | 4.7 | 4.9 | 5.4 | 5.8 | 7.1 | 19 | 19 | 20 | 22 | 23 |
| C03DA | Aldosterone antagonists | 0.7 | 0.7 | 0.8 | 0.9 | 1.0 | 2 | 2 | 2 | 2 | 2 |
| C03DB | Other potassium-sparing agents | 0.1 | 0.1 | 0.1 | 0.1 | 0.1 | 0 | 0 | 0 | 0 | 0 |
| C03XA | Vasopressin antagonists | 0.0 | 0.0 | 0.0 | 0.0 | 0.0 | 1 | 1 | 1 | 1 | 4 |
| C07AA | Beta blocking agents, non-selective | 0.5 | 0.5 | 0.5 | 0.6 | 0.6 | 6 | 6 | 7 | 8 | 8 |
| C07AB | Beta blocking agents, selective | 4.9 | 5.0 | 5.2 | 5.3 | 5.4 | 50 | 48 | 46 | 45 | 44 |
| C07AG | Alpha and beta blocking agents | 2.5 | 2.6 | 2.7 | 2.9 | 3.2 | 66 | 64 | 64 | 67 | 68 |
| C08CA | Dihydropyridine derivatives | 35.5 | 37.0 | 38.9 | 39.8 | 43.9 | 378 | 371 | 370 | 359 | 362 |
| C08DA | Phenylalkylamine derivatives | 0.3 | 0.3 | 0.3 | 0.3 | 0.3 | 4 | 5 | 5 | 5 | 5 |
| C08DB | Benzothiazepine derivatives | 1.1 | 1.2 | 1.3 | 1.3 | 1.2 | 15 | 15 | 15 | 13 | 11 |
| C09AA | ACE inhibitors, plain | 5.8 | 5.5 | 5.3 | 5.2 | 5.2 | 64 | 57 | 52 | 47 | 44 |
| C09CA | Angiotensin II receptor blockers (ARBs), plain | 28.0 | 30.6 | 33.3 | 35.5 | 40.6 | 444 | 455 | 483 | 502 | 539 |
| **Antihypertensive drug combinations** | | | | | | | | | | | |
| C03EA | Low-ceiling diuretics and potassium-sparing agents | - | - | - | - | - | 0 | 0 | 0 | 0 | 0 |
| C07BB | Beta blocking agents, selective, and thiazides | 0.3 | 0.3 | 0.2 | 0.2 | 0.2 | 4 | 4 | 4 | 3 | 3 |
| C07CB | Beta blocking agents, selective, and other diuretics | 1.1 | 1.0 | 0.9 | 0.8 | 0.8 | 7 | 6 | 5 | 5 | 4 |
| C07FB | Beta blocking agents and calcium channel blockers | 0.0 | 0.0 | 0.0 | 0.0 | 0.0 | 0 | 0 | 0 | 0 | 0 |
| C09BA | ACE inhibitors and diuretics | 0.1 | 0.1 | 0.1 | 0.1 | 0.1 | 1 | 1 | 1 | 1 | 1 |
| C09BB | ACE inhibitors and calcium channel blockers | 0.4 | 0.4 | 0.3 | 0.3 | 0.3 | 8 | 7 | 6 | 5 | 5 |
| C09DA | Angiotensin II receptor blockers (ARBs) and diuretics | 48.8 | 49.3 | 50.7 | 51.8 | 52.7 | 716 | 702 | 702 | 696 | 679 |
| C09DB | Angiotensin II receptor blockers (ARBs) and calcium channel blockers | 42.4 | 54.9 | 70.0 | 80.0 | 90.8 | 632 | 797 | 1001 | 1088 | 1177 |
| C09DX | Angiotensin II receptor blockers (ARBs), other combinations | 3.1 | 4.0 | 5.2 | 8.6 | 11.4 | 49 | 59 | 74 | 133 | 166 |
| **Antihyperlipidemic drugs, plain** | | | | | | | | | | | |
| C10AA | HMG CoA reductase inhibitors | 29.2 | 35.0 | 39.9 | 45.8 | 52.1 | 1,141 | 1,259 | 1,366 | 1,491 | 1,593 |
| C10AB | Fibrates | 2.0 | 2.2 | 2.6 | 3.1 | 3.6 | 53 | 57 | 66 | 76 | 89 |
| C10AC | Bile acid sequestrants | 0.0 | 0.0 | 0.0 | 0.0 | 0.0 | 1 | 1 | 1 | 1 | 1 |
| C10AD | Nicotinic acid and derivatives | 0.1 | 0.0 | 0.0 | 0.0 | 0.0 | 3 | 3 | 1 | 1 | 1 |
| C10AX | Other lipid modifying agents | 0.1 | 0.1 | 0.1 | 0.2 | 0.3 | 5 | 5 | 5 | 8 | 11 |
| **Antihyperlipidemic drug combinations** | | | | | | | | | | | |
| C10BA | Combinations of various lipid modifying agents | 2.9 | 8.4 | 14.6 | 20.4 | 27.7 | 96 | 315 | 528 | 703 | 912 |
| C10BX | Lipid modifying agents in combination with other drugs | 7.0 | 10.4 | 13.2 | 15.4 | 18.8 | 175 | 249 | 300 | 333 | 383 |

The JNC8 guideline and Korean Society of Hypertension recommend combination therapy of ARB or ACE inhibitor + CCB, ARB or ACE inhibitor + diuretics, and CCB + diuretics [1, 21]. The Korean Society of Lipid and Atherosclerosis also recommended combination therapy of statin and other lipid-modifying agents in their guideline for the management of dyslipidemia. This result is consistent with the recommendation of FDC use to reach the target.

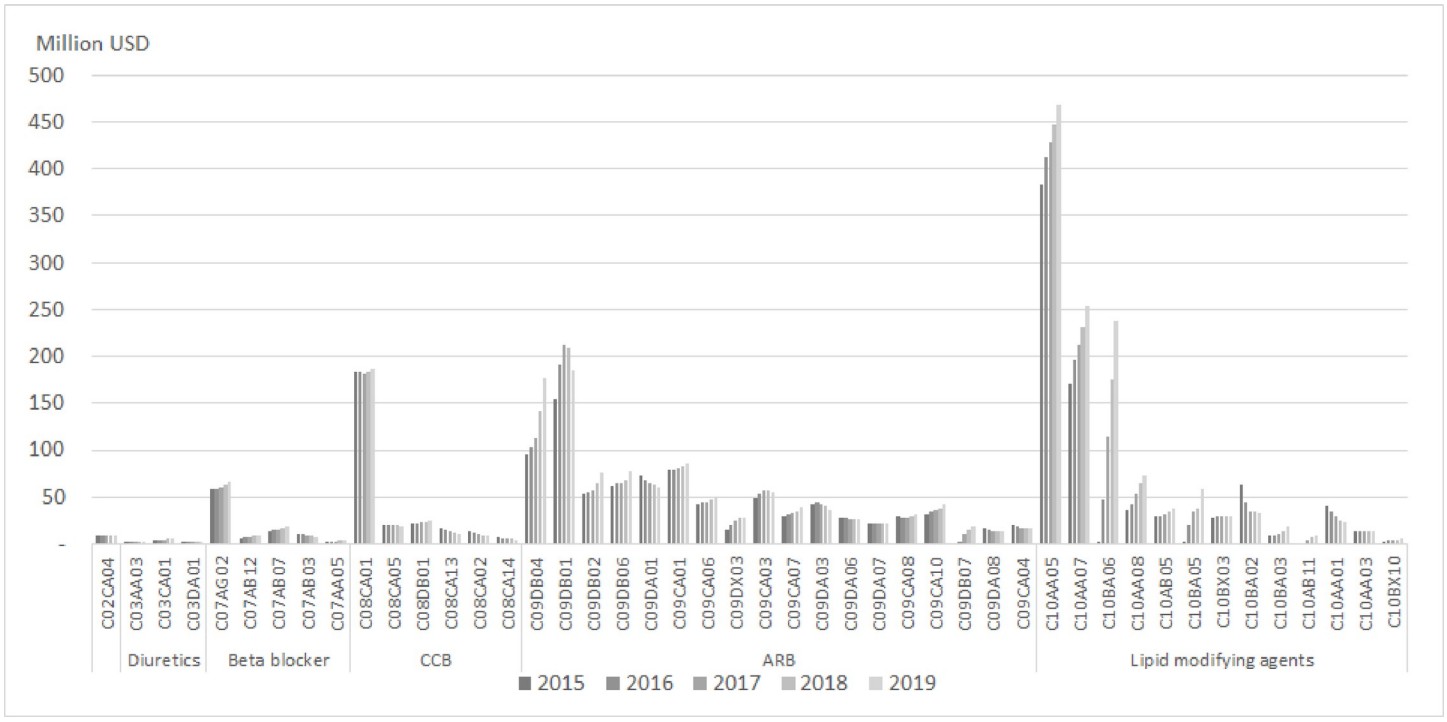

**Fig 1. Drug spending on antihypertensive and antihyperlipidemic drugs by ingredients.** Note. C02CA04: doxazosin, C03AA03: hydrochlorothiazide, C03CA01: furosemide, C03DA01: spironolactone, C07AG02: carvedilol, C07AB12: nebivolol, C07AB07: bisoprolol, C07AB03: atenolol, C07AA05: betaxolol, C08CA01: amlodipine, C08CA05: nifedipine, C08DB01: diltiazem, C08CA13: lercanidipine, C08CA02: felodipine, C08CA14: cilnidipine, C09DB04: telmisartan+amlodipine, C09DB01: valsartan +amlodipine, C09DB02: olmesartan medoxomil+amlodipine, C09DB06: losartan+amlodipine, C09DA01: losartan+diuretics, C09CA01: losartan, C09CA06: candesartan, C09DX03: olmesartan, edoxomil, amlodipine+hydrochlorothiazide, C09CA03: valsartan, C09CA07: telmisartan, C09DA03: valsartan+diuretics, C09DA06: candesartan +diuretics, C09DA07: telmisartan+diuretics, C09CA08: olmesartan medoxomil+diuretics, C09CA10: fimasartan, C09DB07: candesartan+amlodipine, C09DA08: olmesartan medoxomil+diuretics, C09CA04: irbesartan, C10AA05: atorvastatin, C10AA07: rosuvastatin, C10BA06: rosuvastatin+ezetimibe, C10AA08: pitavastatin, C10AB05: fenofibrate, C10BA05:atorvastatin+ezetimibe, C10BX03: atorvastatin+amlodipine, C10BA02: simvastatin+ezetimibe, C10BA03: pravastatin+fenofibrate, C10AB11: choline fenofibrate, C10AA01: simvastatin, C10AA03: pravastatin, C10BX10: rosuvastatin+valsartan.

According to several meta-analysis studies, the FDC treatment is associated with a significant improvement in adherence and persistence in comparison with single therapy or free-equivalent combined therapies (FECs), thereby increase of BP control effect [22, 23], so these strategies have been preferred compared with conventional therapies [24].

To our knowledge, this is first study that conducted an analysis of the pharmaceutical market including single pills and FDCs because there are limited real-world data on the amount of FDCs. Second, considering recall and withdrawal of valsartan due to the carcinogen nitrosodimethylamine (NDMA) in August 2018, this result showed the corresponding change in therapeutic classes. Lastly, most existing studies were limited to a single region, whereas we used a nationwide population dataset to obtain representative results. However, the patterns of clinical practice are different from those reported in other studies; in particular, the use of CCBs was found to be somewhat high in the Korean antihypertensive drug market. Thus, future research investigating physicians' and patients' preferences for antihypertensive drugs might clarify this finding.

Our study has several limitations. First, our analysis used an administrative dataset; thus, information was lacking on the clinical factors associated with therapy selection. For example, the combination of beta-blockers and diuretics might increase the risk of incidence of diabetes mellitus. Because we could not distinguish patients' conditions, we only analyzed the amount of drug consumption and spending. Like other administrative databases, the Korean health

insurance claims data are the basis for payment, resulting in inherent limitations, including incompleteness and inaccuracy in complicated diagnoses. Second, when calculating drug spending, preparation fees by pharmacists were not included in this study.

Despite these limitations, this is the first study to investigate trends in single pill versus fixed-dose combinations in antihypertensive and antihyperlipidemic drugs using real-world data at the national level. In conclusion, consistent with the increased prevalence of hypertension and dyslipidemia due to population aging, the consumption and drug spending of antihypertensive and antihyperlipidemic fixed-dose drugs was higher in older adults, and a marked increase in FDCs was found among older adults, driven by ARBs + CCBs as antihypertensive drugs. Special attention should be paid to real-world evidence on the health outcome of FDCs in terms of improved medication adherence and control of blood pressure or lipid levels in older adults.

## Author Contributions

**Conceptualization:** Dong-Sook Kim.

**Data curation:** Yujin Lee, Jihye Shin.

**Formal analysis:** Jihye Shin.

**Writing – original draft:** Yujin Lee, Dong-Sook Kim.

**Writing – review & editing:** Yujeong Kim, Dong-Sook Kim.

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
