## [Decision Letter · Decision Letter 0]

12 Aug 2021

PONE-D-21-21296

Consumption of single products versus fixed-dose combination medicines for hypertension and hyperlipidemia during 2015-2019 in South Korea

PLOS ONE

Dear Dr. Kim,

Thank you for submitting your manuscript to PLOS ONE. After careful consideration, we feel that it has merit but does not fully meet PLOS ONE’s publication criteria as it currently stands. Therefore, we invite you to submit a revised version of the manuscript that addresses the points raised during the review process.

The manuscript has to be revised according to the Reviewers' suggestion.

See the comments carefully and respond them appropriately.

We look forward to receiving your revised manuscript.

Kind regards,

Masaki Mogi

Academic Editor

PLOS ONE

Journal Requirements:

“No”

“No”

Reviewers' comments:

Reviewer's Responses to Questions

**Comments to the Author**

1. Is the manuscript technically sound, and do the data support the conclusions?

Reviewer #1: Yes

Reviewer #2: Yes

2. Has the statistical analysis been performed appropriately and rigorously? 

Reviewer #1: Yes

Reviewer #2: Yes

3. Have the authors made all data underlying the findings in their manuscript fully available?

Reviewer #1: Yes

Reviewer #2: No

4. Is the manuscript presented in an intelligible fashion and written in standard English?

Reviewer #1: Yes

Reviewer #2: Yes

5. Review Comments to the Author

Reviewer #1: The manuscript is well written and clear to the reader.

Only two remarks:

• In "Use by drug classification" chapter, paragraph 5, line 2, after "...carvedilol (4.8 DIDs, 67 million USD", it needs a parenthesis ")";

• In "Discussion" chapter, paragraph 1, line 9, it is "...followed by C08 (CCB), with 70.7 DIDs".

Reviewer #2: The present study is a retrospective study based on the Korean national health insurance claims database, describing the trends in consumption of fixed-dose combinations of lipid-lowering and antihypertensive drugs. The study is of interest and the following comments should be taken into account to improve its quality.

-The available data regarding triple combinations (e.g. 2 antihypertensive drugs+statin or 1 antihypertensive drug+statin+ezetimibe) should be reported.

-The results of large meta-analyses in the field should be discussed.

-Pharmacokinetic/pharmacodynamic data regarding potential interactions should be elaborated in the Discussion section.

-Figure 1, despite being comprehensive, is complicated. An additional figure is suggested showing the consumption trends of the most common antihypertensive drugs and combinations, presenting the name of the class (e.g. Diuretics) and not the drug codes in order to be easily interpretable by clinicians.

6. PLOS authors have the option to publish the peer review history of their article (what does this mean?). If published, this will include your full peer review and any attached files.

Reviewer #1: **Yes: **Artur Mendes Moura

Reviewer #2: No

---

## [Author Response · Author response to Decision Letter 0]

10 Oct 2021

Consumption of single products versus fixed-dose combination medicines for hypertension and hyperlipidemia during 2015-2019 in South Korea

Response to reviews’ comments

Dear Masaki Mogi

We appreciate the opportunity to revise our manuscript (manuscript ID PONE-D-21-21296), based on your comments in an e-mail that was dated August 13, 2021. The manuscript has been revised based on the comments that were provided. 

We have enclosed the revised manuscript with the changes highlighted as well as a response to the reviewers comment (below). Also, we revised the reference list.

Thank you for your ongoing consideration, and we look forward to hearing from you. 

Sincerely,

Dong-Sook Kim

 

Response to the reviewer #1’s comment

We would like to thank the reviewer for improving the manuscript.

The manuscript is well written and clear to the reader.

Only two remarks:

In "Use by drug classification" chapter, paragraph 5, line 2, after "...carvedilol (4.8 DIDs, 67 million USD", it needs a parenthesis ")";

In "Discussion" chapter, paragraph 1, line 9, it is "...followed by C08 (CCB), with 70.7 DIDs".

Author’s Response: Thank you for the time and effort you invested in thoroughly reviewing this manuscript. According to your comment, we revised. 

 

Response to the reviewer #2’s comment

We would like to thank the reviewer for improving the manuscript.

The present study is a retrospective study based on the Korean national health insurance claims database, describing the trends in consumption of fixed-dose combinations of lipid-lowering and antihypertensive drugs. The study is of interest and the following comments should be taken into account to improve its quality.

-The available data regarding triple combinations (e.g. 2 antihypertensive drugs+statin or 1 antihypertensive drug+statin+ezetimibe) should be reported.

Author’s Response: Thank you for your comment.

The unit of analysis of this study was to medicine. We analyzed the characteristics of patients who use fixed dose combination (FDC). However, multiple use of hypertension and hyperlipidemia combinations in the patient unit was not analyzed. 

-The results of large meta-analyses in the field should be discussed.

-Pharmacokinetic/pharmacodynamic data regarding potential interactions should be elaborated in the Discussion section.

Author’s Response: Thank you for these helpful comments. However pharmaceutical company must submit approval data to Food and Drug Administration, so doctors might prefer to prescribe fixed dose combination (FDC) of hypertensive drugs or hyperlipidemia rather than to combine each drug. Therefore, we did not consider regarding potential interactions because this study examined the pattern or trend of combination drugs. Also, we mentioned the large meta-analyses of fixed-dose combination drugs and changed the Discussion section as follows: 

According to several meta-analysis studies, the FDC treatment is associated with a significant improvement in adherence and persistence in comparison with single therapy or FECs, thereby increase of BP control effect, so these strategies have been preferred compared with conventional therapies.

-Figure 1, despite being comprehensive, is complicated. An additional figure is suggested showing the consumption trends of the most common antihypertensive drugs and combinations, presenting the name of the class (e.g. Diuretics) and not the drug codes in order to be easily interpretable by clinicians.

Author’s Response: Thank you for your suggestion. According to your comment, we revised.

---

## [Decision Letter · Decision Letter 1]

20 Oct 2021

Consumption of single products versus fixed-dose combination medicines for hypertension and hyperlipidemia during 2015-2019 in South Korea

PONE-D-21-21296R1

Dear Dr. Kim,

We’re pleased to inform you that your manuscript has been judged scientifically suitable for publication and will be formally accepted for publication once it meets all outstanding technical requirements.

Kind regards,

Masaki Mogi

Academic Editor

PLOS ONE

Additional Editor Comments (optional):

No further comment.

Reviewers' comments:

Reviewer's Responses to Questions

**Comments to the Author**

1. If the authors have adequately addressed your comments raised in a previous round of review and you feel that this manuscript is now acceptable for publication, you may indicate that here to bypass the “Comments to the Author” section, enter your conflict of interest statement in the “Confidential to Editor” section, and submit your "Accept" recommendation.

Reviewer #1: All comments have been addressed

Reviewer #2: All comments have been addressed

2. Is the manuscript technically sound, and do the data support the conclusions?

Reviewer #1: Yes

Reviewer #2: Yes

3. Has the statistical analysis been performed appropriately and rigorously? 

Reviewer #1: Yes

Reviewer #2: Yes

4. Have the authors made all data underlying the findings in their manuscript fully available?

Reviewer #1: Yes

Reviewer #2: (No Response)

5. Is the manuscript presented in an intelligible fashion and written in standard English?

Reviewer #1: Yes

Reviewer #2: Yes

6. Review Comments to the Author

Reviewer #1: (No Response)

Reviewer #2: The authors have adequately revised their manuscript; therefore, it can be accepted for publication.

7. PLOS authors have the option to publish the peer review history of their article (what does this mean?). If published, this will include your full peer review and any attached files.

Reviewer #1: **Yes: **Artur Mendes Moura

Reviewer #2: No

---

## [Editor Report · Acceptance letter]

6 Dec 2021

PONE-D-21-21296R1 

Consumption of single products versus fixed-dose combination medicines for hypertension and hyperlipidemia during 2015-2019 in South Korea 

Dear Dr. Kim:

I'm pleased to inform you that your manuscript has been deemed suitable for publication in PLOS ONE. Congratulations! Your manuscript is now with our production department. 

Kind regards, 

on behalf of

Dr. Masaki Mogi 

Academic Editor

PLOS ONE